# Ligand Modulation on the Various Structures of Three Zinc(II)-Based Coordination Polymers for Antibiotics Degradation

**DOI:** 10.3390/molecules28072933

**Published:** 2023-03-24

**Authors:** Min Xiong, Ying-Gui Xia, Lu Lu, Jun Wang, A. Mohanty, Yu Wu, Hiroshi Sakiyama, Mohd. Muddassir, Ying Pan

**Affiliations:** 1School of Chemistry and Environmental Engineering, Sichuan University of Science & Engineering, Zigong 643000, Chinascwangjun2011@126.com (J.W.);; 2Department of Chemistry, Berhampur University, Berhampur 760007, India; 3Department of Science, Faculty of Science, Yamagata University, 1-4-12 Kojirakawa, Yamagata 990-8560, Japan; 4Department of Chemistry, College of Sciences, King Saud University, Riyadh 11451, Saudi Arabia; 5The First Dongguan Affiliated Hospital, Guangdong Medical University, Dongguan 523808, China

**Keywords:** antibiotic, Zn(II), coordination polymer, degradation

## Abstract

The efficient removal of organic contaminants from wastewater is, nowadays, a prominent area of study due to its biological as well as environmental significance. Antibiotics are now found in wastewater because of their high use, which has become a source of aquatic pollution. These antibiotics have dangerous implications for people’s health. Hence, effective pharmaceutical removal from wastewater and contaminated water bodies, especially the removal of antibiotics, is of major interest to global research organizations. This is why it is necessary to investigate this class of toxic material in wastewater discharge. We synthesized three different coordination polymers (CPs) in the presence of various assistant carboxylate linkers, namely, [Zn(Hbtc)(dip)]_n_ (**1**), [Zn_4_(1,2-bdc)_4_(dip)_4_]_n_ (**2**), and [Zn(1,4-bdc)(dip)]_n_ (**3**) (3,5-di(1H-imidazol-1-yl)pyridine = dip, 1,3,5-benzenetricarboxylic acid = H_3_btc, 1,2-benzenedicarboxylic acid = 1,2-H_2_bdc, and 1,4-benzendicarboxylic acid = 1,4-bdc). These CPs were characterized by using different techniques, including single-crystal X-ray diffraction. The structural studies demonstrated that in **2**, there are four Zn(II) centers and both centers are in different coordination environments (Zn2 has distorted tetrahedral geometry, whereas Zn1, Zn3, and Zn4 have square pyramidal geometry). Hirshfeld surfaces analysis revealed that different types of intermolecular interactions (C⋯C, H⋯C, H⋯H, O⋯C, N⋯H, and O⋯H) are present in the synthesized CPs. We examined the different antibiotics, such as metronidazole (MDZ), nitrofurazone (NFZ), dimetridazole (DTZ), sulfasalazine(SLA), and oxytetracycline (OXY), degradation behaviors of the synthesized CPs, which showed remarkable degradation efficiency. **1** showed photocatalytic behavior toward the NFZ antibiotic in an aqueous media. This study also showed that these catalysts are stable and reusable under mild conditions.

## 1. Introduction

Various harmful and persistent organic contaminants in water pose a serious threat to ecological security, making water pollution now one of the most serious worldwide environmental problems [1,2,3,4]. Moreover, seriously hazardous substances for the environment, such as industrial waste, dyes, organic pollutants, and pharmaceuticals, are discharged into water sources. Antibiotics are widely utilized in industries, including in the treatment of bacterial illnesses in people and as livestock feed additives, as well as in both human and veterinary medicine. Abuse of antibiotics, especially nitro-based antibiotics, such as metronidazole (MDZ) [5] and dimetridazole (DTZ) [6], releases antibiotic residues into the environment, which are toxic and hazardous to both human and animal health. It is, therefore, critical to develop a method for decomposing such substances in the ecosystem that is effective, sensitive, stable, and benign [7]. Several advanced methods have been employed in order to decompose these antibiotics; however, these methods involve professional operators and have the drawback of being expensive [8]. Significantly, catalysis is deemed an efficient process that shows substantial benefits in extracting organic reaction and degradation pollutants [9,10,11,12,13]. It is crucial to design an effective and stable photocatalyst in order to improve the effectiveness of the photocatalytic process of removing contaminants of antibiotic substances.

Coordination polymers (CPs) are porous crystalline resources that have received considerable attention in the last two decades due to their intriguing assemblies with cationic central parts connected by specific multidentate organic ligands [14,15,16,17]. CPs, unlike some other porous substances such as zeolites, are flexible in nature and can be produced under controlled conditions with high specificity and selectivity. Because of the richness of this coordination chemistry, their optoelectric properties can be modified by altering their structural composition. Due to the implementation of the organic linkers, CPs can also selectively adsorb organic molecules. Adsorption interactions are generally formed by covalent bonding, hydrogen bonding, dative bonding, Van der Waals forces, and π–π interaction interactions.

These materials have demonstrated excellent potential employment in multiple aspects, for example, sensing [18,19,20,21,22], photocatalysis [23,24,25], heterogeneous catalysis [26,27,28,29], gas storage [30,31,32,33], and drug delivery [34,35,36]. They are well known for their extensive structural tunability, high porosity, unique physicochemical properties, and light reactivity. CPs have recently been developed as novel photocatalytic materials for organic pollutant degradation, organic transformation, and atmospheric CO_2_ reduction [37,38,39,40,41,42,43,44,45,46]. To decontaminate organic pollutants via physical adsorption or chemical degradation, visible-light-induced photodegradation of organic pollutants is thoughtfully considered a green, economical, and realistic method for environmental rehabilitation.

The value of these photocatalysts has increased due to fast-growing experimentation on environmentally friendly CPs through the use of environmentally friendly solvents, ligands, and synthesis techniques in a sustainable manner. This allows us to create completely green water treatment. Furthermore, various functional groups in organic ligands and metal nodes act as adsorption centers for different kinds of organic pollutants.

Recently, Wang et al. [47] employed Bi_2_S_3_/MOF-808 nanocomposites with high stability for the photocatalytic degradation of antibiotic tetracyclines. Chen et al. [48] found that the photocatalytic reaction performance of tetracycline (TC) reaches 87.03%, and the rate of degradation of chlortetracycline (CTC) reaches 78.91% in 60 min using the optimal Ti-MOF-derived materials. Zhang et al. [49] found that Fe_3_O_4_/MIL-101(Fe) demonstrates excellent OXY decolorization efficiency (87.1%, 1 h), and it exhibited remarkable extraction efficiency at OXY concentrations of 30 to 70 mg L^−1^ and pH values of 3 to 9. It was further confirmed that these ˙SO_4_^2−^ and ˙OH radicals had an essential part in the OXY decolorization reaction. Moreover, the co-precipitation approach has utilized BiOBr/UiO-66 to degrade atrazine due to its outstanding chemical and structural stability [50].

In this report, assembled from this V-shaped dip ligand (which includes both pyridine and imidazole groups) and different carboxylate linkers, three Zn-containing frameworks, namely, [Zn(Hbtc)(dip)]_n_ (**1**), [Zn_4_(1,2-bdc)_4_(dip)_4_]_n_ (**2**), and [Zn(1,4-bdc)(dip)]_n_ (**3**) (3,5-di(1H-imidazol-1-yl)pyridine = dip, 1,3,5-benzenetricarboxylic acid = H_3_btc, 1,2-benzenedicarboxylic acid = 1,2-H_2_bdc, and 1,4-benzendicarboxylic acid = 1,4-bdc), were solvothermally synthesized (Figure 1). This inventive research shows that when in solution, the organic linker of the CPs frequently produces certain changes that result in the release of electrons from the stimulated linker molecule by the organic linker molecule.

This study examines the use of CPs as catalysts in the photocatalytic degradation of various water pollutants. They exhibit interesting various structures due to the different assistant carboxylates used in the reaction. The degradation of antibiotics, such as metronidazole (MDZ), nitrofurazone (NFZ), dimetridazole (DTZ), sulfasalazine (SLA), and oxytetracycline (OXY), in an aqueous media are investigated in detail.

## 2. Results and Discussion

### 2.1. Crystal Structure of [Zn(Hbtc)(dip)]_n_ (**1**)

Single-crystal X-ray analysis of **1** revealed that it crystallizes in the triclinic system and *P*-1 space group (Appendix A). In **1**, the coordination geometry around the Zn(II) can be described as distorted tetrahedral{ZnN_2_O_2_} (Figure 1a) with the two coordination sites occupied by two oxygen atoms of carboxylate groups from two different Hbtc ligands and two imidazole nitrogen atoms from two different dip ligands. The Zn−N distances are in the 1.991(2) and 2.017(2) Å range; the Zn−O distances are 1.971(2) and 2.016(2) Å (Appendix A). The O···Zn···O angle is almost perpendicular (97.27°) (Appendix A). The asymmetric unit of **1** contains one atom of Zn(II), one Hbtc molecule, and one dip. An intriguing secondary building unit (SBU)-based 3D porous network development is shown by the structural analysis. The dip molecules form the typical 1D helical chain of extended-SBU, while the acid compounds connect these ext-SBUs at the different metal centers to form a 2D structure (Figure 1b). The adjacent chains are connected by strong classical hydrogen bonds of the carboxylic acid, and π−π interactions play a significant role in the self-assembly of these close-packed 2D nets, allowing the structures to furnish their 3D frameworks [51]. The uncoordinated oxygen atom of the carboxylate group in btc is involved in the formation of the C–H⋯O (2.400 Å) interaction between the carboxylate group and the pyridyl fragment of the molecule; similarly, the formation of the C–H⋯O (2.280 Å) interaction between the imidazole group and the carboxylate group stabilizes the framework.

### 2.2. Crystal Structure of [Zn_4_(1,2-bdc)_4_(dip)_4_]_n_ (**2**)

The asymmetric unit of **2** contains four crystallographically independent Zn(II) centers, four 1,2-bdc ligands, and four dip ligands. Zn2 is coordinated by two monodentate oxygen atoms from two 1,2-bdc ligands and two nitrogen atoms from two dip ligands, having the formation of distorted tetrahedral geometry (Figure 2a), while Zn1, Zn3, and Zn4 are coordinated by three oxygen atoms (one monodentate and one bidentate mode from 1,2-bdc ligands) and two nitrogen atoms from dip ligands, resulting in the formation of square pyramidal geometry (Figure 2a). The Zn−O distances are in the range of 1.972(2)−2.579(2) Å (Appendix A). Due to the neighboring benzene and pyridyl rings in the dip being distinct, the nets exhibit π−π stacking interactions, and the dip ligand is long, a two-fold interpenetrating network is constructed to reduce the pore space for structural stability Figure 2b. The 1,2-bdc ligand connects to the metal center, having a distance of 6.179. Similarly, the dip ligand connects to the Zn metal center, having distances of 13.161 and 12.901 Å (Figure 2c). It is important to mention here that the distance between the two metal centers in both CPs is dependent on the nature of the ligands involved in bridging.

A 2D structure was formed by further connecting the dinuclear “Zn” units with 1,2-bdc ligands, which was then constructed into 3D self-penetrating networks by the coupling of dip ligands, as shown in Figure 2d.

### 2.3. Crystal Structure of [Zn(1,4-bdc)(dip)]_n_ (**3**)

The crystallography structure of **3** shows that there is one Zn(II) ion, one 1,4-bdc ligand, and one dip ligand in the asymmetric unit. The Zn1 ion possesses a distorted tetrahedral structure composed of two nitrogen atoms from two dip ligands (Zn−N, 2.013(4) Å) (Appendix A) and two oxygen atoms from two 1,4-bdc ligands (Zn−O, 1.956–1.984 Å) (Figure 3a). The three-dimensional structure is produced by the 1,4-bdc ligands and dip ligands linking the adjacent Zn ions together with 1D (the hydrogen bonds of the carboxylic acid and π−π stacking play a major role in the self-assembly of these 1D polymeric networks) channels along the a-axis. There are various hydrogen bonds between the 2D layers. The adjoining 2D layers, in particular, have π−π stacking interactions with a centroid–centroid distance of 3.457 between the aromatic nuclei of the dip ligands. When the 1,4-bdc ligands are connected to the Zn metal center along the a-axis, the distance is 11.048; similarly, when the 1,4-bdc ligands connect to the Zn metal center along the b-axis, the distance changes from 11.048 to 10.958 due to the orientation of ligand geometry. The uncoordinated oxygen atom of the carboxylate group in 1,4-bdc is involved in the formation of the C–H⋯O (2.423 Å) interaction between the carboxylate group and the pyridyl molecule; similarly, the formation of the C–H⋯O (2.365 Å) interaction between the imidazole group and the carboxylate group stabilizes the framework. As a result, CPs3 is a 3D supramolecular polymer composed of hydrogen bonds and stacking interactions [52] (Figure 3d).

### 2.4. TGAs and PXRD Patterns

The thermal stabilities of **1**−**3** were determined in the temperature range of room temperature (RT) to 800 °C. The TG experiment was conducted in a N_2_ environment on the crystalline samples of **1**–**3**. The temperature was increased at 10 °C/min from 40 to 800 °C (Appendix A). According to the result analyses, the first step of decomposition can be attributed to the loss of the uncoordinated/coordinated solvent molecule within a temperature range of 100–250 °C. In the second step, weight loss corresponds to the collapse of the framework. The decomposition of the uncoordinated/coordinated carboxylate anion occurred at 180–390 °C. Initially, the pyridine moiety started decomposing and continued up to 300 °C. In the end, all the CPs form M-O as the product at 802 °C to 830 °C for **1**–**3**.

Before the degradation measurements, at room temperature, solid-state PXRD measurements were performed to verify the bulk purity of the as-synthesized powder samples of **1**−**3**. The diffraction peaks of **1**−**3** in the crushed crystalline state were compared and matched with the simulated one to confirm their phase purities. As shown in (Appendix A), the PXRD pattern at room temperature also shows that the crystal structures are similar to the simulated ones, revealing the phase purity of the samples. Importantly, the preferred orientation of the lattice face may be the cause of the peak discrepancies in intensity between the simulated and experimental patterns.

The Fourier-transform infrared (FT-IR) method is useful for identifying organic materials and functional groups in a variety of substances. FTIR spectra were obtained for **1**–**3** (Appendix A). The spectrum for CPs **1**–**3** exhibits 3000–3600 cm^−1^ for the C-H vibration. C=O stretching, which is displaced due to coordination with the metallic center, is responsible for the strong absorption band at 1576 cm^−1^. The peaks in the FTIR spectra of **1**–**3** at 3200 to 2800 cm^−1^ are attributed to aromatic C-H stretching and aliphatic C-H stretching in the imidazole, respectively [53]. Strong peaks at 1601 cm^−1^, 1409 cm^−1^, and 1384 cm^−1^ should be due to the vibrational absorption of the carboxylate group in the ligand. The stretching of the coordination Zn-N is attributed to the absorption band at 421–1000 cm^−1^. Thus, Zn-CPs were formed based on the results of the XRD and FTIR analyses.

### 2.5. Photocatalytic Degradation of Antibiotics

Antibiotics have been widely utilized to treat infections in both humans and animals, and their use is gradually rising in the worldwide industry. Most antibiotics are poorly absorbed and continually released into the environment. Several chemical and physical techniques, including improved oxidation, electrochemistry, membrane filtering, and adsorption, have been developed for the removal of pharmaceutical antibiotics. Antibiotics are ionic, heterocyclic organic hazardous materials that are very stable against oxidation reactions and have been widely used to protect human health and promote the growth of flora and fauna [54,55]. The band gaps (Eg) of CPs **1**–**3** were calculated using the diffuse reflectance (DR) UV-vis data. The subsequent values were determined as the cross-site between the x-axis, and the lines were inferred by the straight line position of the absorption boundary. The *E*_g_ values obtained were 3.76, 3.82, and 3.93 eV (Appendix A).

A photocatalytic degradation experiment on various antibiotics was carried out to evaluate the photocatalytic activity of the synthesized compounds under the influence of visible light irradiation. Photocatalysts were first immersed in antibiotic solutions and reacted in the dark for 30 min to reach the equilibrium of adsorption/desorption. Stirring was continued during the photodegradation reaction to maintain the mixture in suspension. At regular intervals, samples were collected and centrifuged to isolate the photocatalyst for analysis. The UV−vis absorption characteristics of antibiotics decreased gradually with time (Figure 4a, Appendix A) in the presence of CP photocatalysts, indicating continuous photocatalytic antibiotic degradation. The C_t_/C_0_ plots of all the antibiotics against reaction time were observed (Figure 4e), which reveals that NFZ underwent the greatest amount of degradation. A kinetics study was carried out to better understand the photocatalytic reaction; this revealed that the experimental data obtained using the Langmuir–Hinshelwood equation fit well with the pseudo-first-order model. Based on the linear plot (Figure 4f), ln(*C*_0_/*C*), and reaction time, the kinetic rate constant, *k*, was calculated [56]. NFZ was used in additional investigations since it had the highest rate of photodecomposition.

### 2.6. Influence of the Dosage of 1

The ideal dosage was investigated in order to prevent excessive consumption of **1**. This study reveals that efficiency increased as dosage percentages were increased, reaching 87% at 20 mg, after which there was a decrease at increasing doses (Figure 4d). This outcome can be explained by the solution’s turbidity and the decreased availability of active sites at very high dosages.

### 2.7. Influence of the NFZ Concentration

The antibiotic’s initial concentration may also have an effect on the photocatalytic performance of the photocatalysts. As a result, it was determined how starting NFZ concentrations (10, 20, and 30 ppm) affected the effectiveness of photodegradation utilizing 1. According to the findings, 76% of NFZ was photodegraded at 10 ppm, while at 20 ppm, NFZ’s photodegradation efficiency increased, and approximately, 87% of the initial NFZ was destroyed after 45 min of irradiation. Then, the percentage of decomposition decreased to 70% when the NFZ concentration increased (to 30 ppm) (Figure 5d).

### 2.8. Research on the Mechanism of Photocatalysis

To determine the active species for the degradation of antibiotics, trapping experiments were performed using different scavengers, such as by adding 1 mM ammonium oxalate (AO, as a quencher of holes, h^+^), 1 mM benzoquinone (BQ, as a quencher of O_2_˙^−^), or 1 mM tertiary butyl alcohol (TBA, as a quencher of ˙OH) [57] (Figure 6d). The degradation of antibiotics did not significantly diminish when AO was added to the reaction, suggesting that h^+^ was not the major active species either. However, the degradation of antibiotics was significantly inhibited by the addition of BQ and TBA, indicating that O_2_˙^−^ and ˙OH played a crucial role [58,59,60,61] in this photocatalytic reaction and confirming that ˙OH was the major active species, as shown in Figure 2.

### 2.9. Recyclability and Stability

Reusability and photostability are important requirements for practical photocatalyst applications. To evaluate the reusability of the synthesized CP catalysts, recycling experiments were carried out. As a result, the CPs’ material was recovered and washed after each run to remove the residual antibiotic for the next photodegradation experiment. After four cycles, the photodegradation activity of the CP catalysts was retained but slightly deactivated (Appendix A). CPs, after recycling experiments, were easily collected by centrifugation. Simultaneously, the collected sample’s PXRD patterns remained unchanged after four cycles, indicating high sensitivity, recyclability, and stability for the degradation of antibiotics (Appendix A).

### 2.10. Hirshfeld Surface Analysis

Crystal Explorer software is generally applied in Hirshfeld surface analysis, especially in the study of intermolecular interactions and exterior molecular environments. The intermolecular interactions of **1**–**3** were analyzed using Hirshfeld surface analysis to evaluate how carboxylates, i.e., H3btc, 1,2-H2bdc, and 1,4-bdc, interact with imidazole in the framework. Hirshfeld surface analysis suggests the possibility of learning more about intermolecular interactions and defining the volume of the promolecule electron density, whereas 2D fingerprint plots reveal the distances of the Hirshfeld surface to the nearest nucleus interior surface (interior, di) and exterior surface (interior, di; exterior, de). The Hirshfeld d_norm_ surface was used for the description of the intermolecular interactions. Dark-red spots act for effective classical interactions in the Hirshfeld surface analysis, and light-red spots act for non-classical interactions or Van der Waals interactions.

For **1**, C···H/H···C and H···H interactions reflect 13.2 and 48.3%, respectively; the amount of which is more than half of the total. The proportions of O ···H/H ···O, C···C, and N···H interactions comprise 30.4%, 5.4%, and 9.1%. The proportion of O···H/H···O interactions is 20.6% of the Hirshfeld surfaces of 1,2-bdc in **2**, whereas in **3**, it is 11.6%, suggesting that imidazole interacts with the O-atoms of the carboxylate groups. The proportions of O···H/H···O interactions are responsible for the formation of a stronger framework. The percentage of non-classical interactions viz. C···H/H···C, C···C, and H···H are 7.1%, 5%, and 38% in **2** and 8.9, 6.7, and 39.2 in **3**. A close investigation of the Hirshfeld surface analysis of these CPs showed that the maximum area of the Hirshfeld surface is comprised of O···H/H···O and H…H types of interactions (Figure 7).

## 3. Experimental Method

### 3.1. Materials

All reagents and solvents were analytical-reagent-grade. The powder X-ray diffraction (PXRD) spectra were obtained by using a Bruker D8 diffractometer with monochromatic (CuKα, λ = 1.540 Å) radiation with a scanning rate of 6°/min and a step size of 0.02°. The structures were solved by direct methods and refined by full-matrix least-squares on F^2^. Structure solution, refining, and data output were performed with the SHELXTL program. Non-hydrogen atoms were refined anisotropically. Images and hydrogen bonding interactions were created in the crystal lattice with DIAMOND-4.6 and MERCURY. Fourier-transform infrared (FT-IR) spectrum in KBr disc was recorded using Nicolet Impact 750 FTIR between 400 and 4000 cm^−1^. Thermogravimetric analysis (TGA) was performed under nitrogen atmosphere from room temperature to 900 °C with a heating rate of 10 °C min^−1^. The Cif files for the structures of CPs **1**–**3** were deposited with the Cambridge Crystallographic Data Centre (CCDC) as deposition nos. **1**–**3** and are 2,190,809, 2,190,811, and 2,190,810.

### 3.2. Sample Preparation

#### 3.2.1. Preparation of [Zn(Hbtc)(dip)]_n_ (**1**)

A mixture solution of Zn(NO_3_)_2_·6H_2_O (0.25 mmol, 0.074 g), bip (0.1 mmol, 0.023 g), H_3_btc (0.1 mmol, 0.021 g), and 6 mL of CH_3_CN/H_2_O (*v/v* = 1:1) was added to 2d NaOH (0.5 mol/L), and the mixture was stirred for 30 min. The mixture was placed in a Teflon-lined reactor and heated at 120 °C for three days. White crystals were obtained after cooling the reaction mixture to room temperature. The crystal was washed with MeOH. The yield was 63% based on Zinc. Elemental anal. calcd. C_20_H_13_N_5_O_6_Zn: C, 49.46; H, 2.56; N, 14.48; Found: C, 50.01; H, 2.53; N, 14.57.

#### 3.2.2. Preparation of [Zn_4_(1,2-bdc)_4_(dip)_4_]_n_ (**2**)

A mixture of bip (0.1 mmol, 0.023 g), 1,2-bdc (0.1 mmol, 0.016 g), Zn(NO_3_)_2_·6H_2_O (0.25 mmol, 0.074 g), and 6 mL of CH_3_CN/H_2_O (*v/v* = 1:1) was combined, and the mixture was stirred for 30 min. The mixture was placed in a Teflon-lined reactor and heated at 120 °C for three days. White crystals were obtained after cooling the reaction mixture to room temperature. The crystal was washed with MeOH. The yield was 70% based on Zinc. Elemental anal. calcd. C_19_H_13_N_5_O_4_Zn: C, 51.78; H, 2.97; N, 15.89 Found: C, 51.99; H, 3.00; N, 15.97.

#### 3.2.3. Preparation of [Zn(1,4-bdc)(dip)]_n_ (**3**)

A mixture of bip (0.1 mmol, 0.023 g), 1,4-bdc (0.1 mmol, 0.016 g), Zn(NO_3_)_2_·6H_2_O (0.25 mmol, 0.074 g), and 6 mL of CH_3_CN/H_2_O (*v/v* = 1:1) was combined, and the mixture was stirred for 30 min. The mixture was placed in a Teflon-lined reactor and heated at 120 °C for three days. White crystals were obtained after cooling the reaction mixture to room temperature. The crystal was washed with MeOH. The yield was 66% based on Zinc. Elemental anal. calcd. C_38_H_26_N_10_O_8_Zn_2_: C, 51.78; H, 2.97; N, 15.89; Found: C, 51.96; H, 3.01; N, 15.96.

### 3.3. Photocatalytic Method

The photocatalytic activities of **1**–**3** (40 mg) were evaluated by the degradation of antibiotic pollutants in the aqueous solution under a UV-400-type photochemical reactor with a 400 W mercury lamp (mean wavelength, 365 nm). An aqueous solution of 50 mL 20 mg/L antibiotics was mixed with 0.01 mmol photocatalyst. The suspension containing antibiotics and photocatalysts was magnetically stirred for 30 min in the dark until adsorption–desorption equilibrium was established. A 3.5 mL sample was extracted at 5 min intervals using 3 mL pipettor and centrifuged to remove the residual catalyst for analysis on a UV–visible spectrophotometer at an absorption wavelength applied to monitor the photocatalytic degradation. In addition, a control experiment was also accomplished with the following reaction conditions: (1) without photocatalyst under UV irradiation; (2) with photocatalyst under UV irradiation in the presence of 2 mL tert-butanol (*t*-BuOH); (3) 2 mL benzoquinone (BQ) was used instead of t-BuOH; (4) 2 mL ammonium oxalate (AO) was used instead of TBA. The degradation efficiency of antibiotics is defined as follows:Degradation efficiency = (C_0_ − C)/C_0_ × 100%
where C_0_ (mg/L) is the initial concentration of dyes, and C (mg/L) is the concentration of dyes at reaction time, t (min).

## 4. Conclusions

In summary, we successfully synthesized three new CPs using a 3,5-di(1H-imidazol-1-yl)pyridine (dip) ligand and different carboxylic acids by using a hydrothermal technique. The results demonstrate that the use of different carboxylates shows that coordination ability, as well as metal centers with varying coordination environments, has a remarkable impact on the structures of the framework. Furthermore, the ligands’ various coordination modes may play an essential part in the development of the coordination framework. The structural studies demonstrate that **2** has four Zn(II) centers, and both centers are in different coordination environments (Zn2 has distorted tetrahedral geometry, whereas Zn1, Zn3, and Zn4 have square pyramidal geometry). These materials work well as functional photocatalysts for the degradation of antibiotics when exposed to visible light. At an apparent rate constant of 0.0320 min^−1^, the photocatalytic degradation efficiency of antibiotics over CPs was found to be over 82% in 45 min, which is similar to some of the best composite photocatalysts.

## Data Availability

Not applicable.

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
