# Peer review of "Ligand Modulation on the Various Structures of Three Zinc(II)-Based Coordination Polymers for Antibiotics Degradation"

_molecules, 2023, doi:10.3390/molecules28072933_

Round 1

Reviewer 1 Report

Manuscript Number: molecules-2270509
Title: Ligand-modulation on the various structures of three zinc(II)-based coordination polymers for antibiotics degradation

Authors: Ming Xiong, Ying-Gui Xia, Lu Lu, Jun Wang *, A Mohanty, Yu Wu, Hiroshi Sakiyama, Mohd. Muddassir, Ying Pan

Three different coordination polymers namely, [Zn(Hbtc)(dip)]n (1), [Zn4(1,2-bdc)4(dip)4]n (2) and [Zn(1,4 bdc)(dip)]n (3) (where 3,5-di(1H-imidazol-1-yl)pyridine = dip, 1,3,5-benzenetricarboxylic acid = H3btc, 1,2-benzenedicarboxylic acid = 1,2-H2bdc and 1,4-benzendicarboxylic acid = 1,4-bdc) were characterized by different techniques including single-crystal X-ray diffraction. The dimensionality of each MOF differ from 1D to 3D. Moreover, the antibiotic degradation behavior of synthesized CPs in an aqueous
media was studied.

This study is interesting, but there are still various significant problems to be solved before the acceptance of this manuscript.

·         It is stated in the abstract “The X-ray studies suggested that the metal centre (?) in all
MOFs are six coordinated although the dimensionality of each MOF differ from 1D to 3D”. This statement contradicts the discussion of the crystal structures of the studied coordination polymers.

·         It is advisable to add an explanation of the abbreviations of the ligands (Hbtc, dip an so on) in the studied complexes in the introduction of the article as well as in reaction scheme 1.

·         The schematic diagram (scheme 1) of the preparation of complexes 1-3 is largely confusing. Instead of showing different types of coordination polymer structures, it is more appropriate to schematically express different types of structures using structural formulas.

·         The abbreviations MOF-1 to MOF-3 used in the description of the figures are not explained in the text.

·         The title of chapter 2.4 (TGAs and PXRD Patterns) does not express the content of the chapter, which also deals with the interpretation of infrared (IR) spectra.

·         The interpretation of thermal (TGAs) and powder (XRD) measurements is only very general.

·         In chapter 2.4, only the IR spectrum of complex 3 are discussed. No IR spectra are listed in the ESI and it is appropriate to add them there.

·         The assignment of the bands in the IR spectrum of complex 3 is controversial. For example, the spectrum for MOF1-3 exhibits 3000–3600 cm-1 for OH of carboxylic acid, however, according to the composition [Zn(1,4-bdc)(dip)]n (3) complex contains only the deprotonated COO group.

·         The level of English is poor. Many typographical and grammatical errors must be corrected.

Notwithstanding the above comments, the study has been competently done and finally the conclusion is of interest to the readership of this journal.

Author Response

Referee 1

This study is interesting, but there are still various significant problems to be solved before the acceptance of this manuscript.

  1. It is stated in the abstract “The X-ray studies suggested that the metal centre (?) in all MOFs are six coordinated although the dimensionality of each MOF differ from 1D to 3D”. This statement contradicts the discussion of the crystal structures of the studied coordination polymers.

Ans: We are very thankful to you Sir for pointing out mistake. We have corrected the coordinate structure in the revised manuscript.

  1. It is advisable to add an explanation of the abbreviations of the ligands (Hbtc, dip an so on) in the studied complexes in the introduction of the article as well as in reaction scheme 1.

Ans: As per your suggestion we have incorporated abbreviations of the ligands in text as well as in scheme in the revised manuscript.

  1. The schematic diagram (scheme 1) of the preparation of complexes 1-3 is largely confusing. Instead of showing different types of coordination polymer structures, it is more appropriate to schematically express different types of structures using structural formulas.

Ans: As per your suggestion we have incorporated structural formulas in scheme in the revised manuscript.

  1. The abbreviations MOF-1 to MOF-3 used in the description of the figures are not explained in the text.

Ans: We are very thankful to you Sir for pointing out mistake. We have corrected the abbreviation i,e MOF-1 is written as 1 and so on

  1. The title of chapter 2.4 (TGAs and PXRD Patterns) does not express the content of the chapter, which also deals with the interpretation of infrared (IR) spectra.

Ans: As per your suggestion we revised the interpretation the TGAs and PXRD Patterns in the revised manuscript.

  1. The interpretation of thermal (TGAs) and powder (XRD) measurements is only very general.

Ans: As per your suggestion we revised the interpretation the TGAs and PXRD Patterns in the revised manuscript.

  1. In chapter 2.4, only the IR spectrum of complex 3 are discussed. No IR spectra are listed in the ESI and it is appropriate to add them there.

Ans: we have incorporated the IR data in the revised manuscript.

  1. The assignment of the bands in the IR spectrum of complex 3 is controversial. For example, the spectrum for MOF1-3 exhibits 3000–3600 cm-1 for OH of carboxylic acid, however, according to the composition [Zn(1,4-bdc)(dip)]n (3) complex contains only the deprotonated COO group.

Ans: The 3000–3400 cm−1 peaks come due to O-H stretching     alcohol because the reaction is carried out in methanol solvent. Again we recollected the data there is no peak in between 3000- 3400 cm−1

  1. The level of English is poor. Many typographical and grammatical errors must be corrected.

Ans: I am sorry for mistake of English in paper and have tried to correct in the revised manuscript.

Reviewer 2 Report

The research work entitled "Ligand-modulation on the various structures of three zinc(II)-based coordination polymers for antibiotics degradation" by M. Xiong et. al. is a nice piece of work. I recommend the article to be publish in "Molecules" subject to following minor corrections:

(i) According to the author, The X-ray studies suggested that the metal centre in all MOFs are six coordinated, however in the crystal structure of 1 and 3 have tetrahedral, while 2 have tetrahedral and five coordinated structure.

(ii) Did the title catalyst have some leaking during photo-degradation experiments?

(iii)  Please verify that the word "NFZdye@10" in Figure 5 is correct.

(iv)    Which solvent was utilized to wash the crystal after photodegradation experiments in terms of recyclability and stability.

(v)    Comparative study of photocatalytic degradation of used antibiotics with other known materials will give a, better insight in the work.

(vi)  Few recent literature to be cited, Polyherdon, 215 (2022) 115693, ACS Omega  2022, 7, 45, 41120  and Cryst. Growth Des. 2022, 22, 12, 7374–7394

Author Response

Referee 2

The research work entitled "Ligand-modulation on the various structures of three zinc(II)-based coordination polymers for antibiotics degradation" by M. Xiong et. al. is a nice piece of work. I recommend the article to be publish in "Molecules" subject to following minor corrections:

(i) According to the author, The X-ray studies suggested that the metal centre in all MOFs are six coordinated, however in the crystal structure of 1 and 3 have tetrahedral, while 2 have tetrahedral and five coordinated structure.

Ans: We are very thankful to you Sir for pointing out mistake. We have corrected the coordinate structure in the revised manuscript.

(ii) Did the title catalyst have some leaking during photo-degradation experiments?

Ans: Thank you for your kind suggestion, we have also confirmed that the full skeleton is stable after the degradation. See the PXRD. Furthermore, we have done the ICP, the Zn content is only 0.365%. Thus, it will not be caused the further pollutants.

(iii)  Please verify that the word "NFZdye@10" in Figure 5 is correct.

Ans: We are very thankful to you Sir for pointing out mistake. We have corrected the data in the revised manuscript.

(iv) Which solvent was utilized to wash the crystal after photodegradation experiments in terms of recyclability and stability?

Ans: we only used the water as solvent in washing work.

(v)    Comparative study of photocatalytic degradation of used antibiotics with other known materials will give a, better insight in the work.

Ans: as per your suggestion we have incorporated the Comparison table of the photocatalytic degradation in Table S3 in the revised manuscript.

(vi)  Few recent literature to be cited, Polyherdon, 215 (2022) 115693, ACS Omega  2022, 7, 45, 41120  and Cryst. Growth Des. 2022, 22, 12, 7374–7394

Ans: As per your suggestion we have cited the references in ref. 53, 61 and 62 in the revised manuscript.

Reviewer 3 Report

The article is of interest to chemists working in the field of supramolecular chemistry, coordination polymers and photocatalysis. Three different zinc coordination polymers in the presence (3,5-di(1H-imidazol-1-yl)pyridine and three different carboxylate linkers were obtained and characterized. The X-ray studies suggested the formation of MOF’s with dimensionality from 1D to 3D. It has been shown that some antibiotics degradation behaviors are observed in the presence of the synthesized coordination polymers.

Some remarks:

1. In the abstract, the authors did not indicate what type antibiotics were studied.

2. Section 2.5 needs to be rewritten. At the beginning of the section there are general words about their prevalence in the environment, then there is a discussion of “The band gap (Eg) of MOF1-3”, and then again general words about antibiotics. It is necessary to justify the choice of the studied antibiotics; it is possible to give their structure formulas SI.

3. Page 5: The phrase “pyridyl molecule” should be written as the “pyridyl part (or fragment) of the molecule”.

4. The phrase «The described Zn-containing structures provide wastewater treatment plants with a biocompatible substrate for the degradation of antibiotics» in the conclusion of the article is not clear. What does the term biocompatible refer to?

5. What is the role of the coordination polymer in photocatalysis in this case? They are not porous. Does the MOF affect the energy gap (band gap), whose value are >3.6–3.9 eV (340–320 nm), which is quite large in itself. Wouldn't it have been easier to use an ordinary complex based on such ligands?

6. Powder XRD of polymers 2 and 3 after photocatalysis has not been measured, are they reliably stable.

7. It would be possible to add to the SI the drawings of the IR Fourier spectra, which are discussed in the article.

8. It’s an error at the page 2 “demo-stratws”

9. It should be noted on the Scheme 1, which structure is 1, which is 2, and which is 3

10. In the text of article (section 2), when crystallographic data are discussed, there are no mentions of Tables 1 and 2 in SI.

11. Page 3 section 2.1. Typo in two places (“corboxylate”).

12. In the section 2.2 Fig. 2 a-c no mentioned

13.  The numbering of complexes 1-3 is not uniform throughout the article - sometimes in bold, sometimes in normal fonts.

14. Make the inscriptions to the figures more clear and detailed, indicating the designations and conditions for experiments.

Author Response

Referee 3

The article is of interest to chemists working in the field of supramolecular chemistry, coordination polymers and photocatalysis. Three different zinc coordination polymers in the presence (3,5-di(1H-imidazol-1-yl)pyridine and three different carboxylate linkers were obtained and characterized. The X-ray studies suggested the formation of MOF’s with dimensionality from 1D to 3D. It has been shown that some antibiotics degradation behaviors are observed in the presence of the synthesized coordination polymers.

Some remarks:

  1. In the abstract, the authors did not indicate what type antibiotics were studied.

Ans: As per your suggestion we have incorporated the type of antibiotics in the abstract in the revised manuscript.

  1. 2. Section 2.5 needs to be rewritten. At the beginning of the section there are general words about their prevalence in the environment, then there is a discussion of “The band gap (Eg) of MOF1-3”, and then again general words about antibiotics. It is necessary to justify the choice of the studied antibiotics; it is possible to give their structure formulas SI.

Ans: As per your suggestion we have rewrite the 2.5 section. Although the composites are expected to perform higher efficiency considering their better photo-sensitiveness and narrower Eg values. Again as per your suggestion we have incorporated the antibiotics structure formula in the SI in the revised manuscript.

  1. Page 5: The phrase “pyridyl molecule” should be written as the “pyridyl part (or fragment) of the molecule”.

Ans: Thank you for your suggestion, we have changed the pyridyl part in the revised manuscript.

  1. The phrase «The described Zn-containing structures provide wastewater treatment plants with a biocompatible substrate for the degradation of antibiotics» in the conclusion of the article is not clear. What does the term biocompatible refer to?

Ans: RBC@Fe3O4 biomimetic nanoparticles, which are highly biocompatible and have great potential for clinical applications. Biomater. Sci., 2021,9, 826-834.

But although we have deleted the line in the conclusion in the revised manuscript.

  1. What is the role of the coordination polymer in photocatalysis in this case? They are not porous. Does the MOF affect the energy gap (band gap), whose value are >3.6–3.9 eV (340–320 nm), which is quite large in itself. Wouldn't it have been easier to use an ordinary complex based on such ligands?

Ans: Yes, this is a really question. We appreciate your suggestion. It may occur easily to use an ordinary complex based on such ligands. The photo-generated holes (h+) as well as .O2- radicals have participated in the degradation of NFZ. This process may be encapsulated by the reaction consisted of multiple stages. We will explore this interesting comment in our next exploration.

  1. Powder XRD of polymers 2 and 3 after photocatalysis has not been measured, are they reliably stable.

Ans: Powder XRD of polymers 2 and 3 after photocatalysis are stable although we have incorporated in the revised manuscript.

  1. It would be possible to add to the SI the drawings of the IR Fourier spectra, which are discussed in the article.

Ans: As per your suggestion we have incorporated the IR Fourier spectra in the revised manuscript.

  1. It’s an error at the page 2 “demo-stratws”

Ans: We are very thankful to you Sir for pointing out mistake. We have corrected the data in the revised manuscript.

  1. It should be noted on the Scheme 1, which structure is 1, which is 2, and which is 3

Ans: We have mention the structure name in the revised manuscript.

  1. In the text of article (section 2), when crystallographic data are discussed, there are no mentions of Tables 1 and 2 in SI.

Ans: We have mention the structure name in the revised manuscript.

  1. Page 3 section 2.1. Typo in two places (“corboxylate”).

Ans: We are very thankful to you Sir for pointing out mistake. We have corrected the data in the revised manuscript.

  1. In the section 2.2 Fig. 2 a-c no mentioned

Ans: We are very thankful to you Sir, we have mentioned the fig. a-c in the revised manuscript.

  1. The numbering of complexes 1-3 is not uniform throughout the article - sometimes in bold, sometimes in normal fonts.

Ans: We are very thankful to you Sir for pointing out mistake. We have corrected the data in the revised manuscript.

  1. Make the inscriptions to the figures more clear and detailed, indicating the designations and conditions for experiments.

Ans: Asper your suggestion we have corrected the figure descriptions in the revised manuscript.

Round 2

Reviewer 1 Report

I consider the answers to my comments to be sufficient.